# Ethanol as a Probe for the Mechanism of Bubble Nucleation in the Diet Coke and Mentos Experiment

**DOI:** 10.3390/molecules26061691

**Published:** 2021-03-17

**Authors:** Thomas S. Kuntzleman, Jacob T. Kuntzleman

**Affiliations:** 1Department of Chemistry, Spring Arbor University, Spring Arbor, MI 49283, USA; 2Department of Physics, Duke University, Durham, NC 27710, USA; jacob.kuntzleman@duke.edu

**Keywords:** bubble nucleation, bubbles, surface science, soda, carbonated beverages, sparkling water, CO_2_, foam

## Abstract

The Diet Coke and Mentos experiment involves dropping Mentos candies into carbonated beverages to produce a fountain. This simple experiment has enjoyed popularity with science teachers and the general public. Studies of the physicochemical processes involved in the generation of the fountain have been largely informed by the physics of bubble nucleation. Herein, we probe the effect of ethanol addition on the Diet Coke and Mentos experiment to explore the impact that beverage surface tension and viscosity have on the heights of fountains achieved. Our results indicate that current descriptions of the effects of surface tension and viscosity are not completely understood. We also extend and apply a previously reported, simplified version of Brunauer–Emmett–Teller theory to investigate kinetic and mechanistic aspects of bubble nucleation on the surface of Mentos candies in carbonated beverages. A combination of this new theory and experiment allows for the estimation that the nucleation sites on the Mentos candy that catalyze degassing are 1–3 μm in size, and that between 50,000 and 300,000 of these sites actively nucleate bubbles on a single Mentos candy. While the methods employed are not highly sophisticated, they have potential to stimulate fresh investigations and insights into bubble nucleation in carbonated beverages.

## 1. Introduction

Dropping Mentos candies into a freshly opened bottle of Diet Coke (or other carbonated beverage) is a popular experiment among science teachers and the general public [1,2,3,4,5,6,7,8,9,10]. The wide-spread fascination with this experiment stems from the fact that it produces impressive results and is extremely easy to carry out. When the candies enter the beverage, they catalyze a sudden degassing of dissolved CO_2_ [3,8,9,10]:


CO_2_ (aq) → CO_2_(g)(1)


The rapid escape of gas forces a jet of bubbly liquid out of the bottle that can reach up to a few meters high. Insight into the physical chemistry involved in fountain formation has been uncovered by studies on the kinetics of degassing and bubble growth in this experiment [6,7,8,9,10]. Further understanding of the relevant processes has been guided by studies on nucleation and bubble growth in Champagne and other carbonated beverages [11,12,13,14,15,16,17,18,19,20]. We now turn to this information.

Carbonated soft drinks (sodas) are sealed under CO_2_ pressures (P_CO_2__) of about 3–5 bar [20,21]. This causes CO_2_ to dissolve into a beverage to a concentration ([CO_2_]) in accordance with Henry’s Law:

[CO_2_] = *K_H_*P_CO_2__(2)
where *K_H_* is the Henry’s Law constant for CO_2_ dissolving in water. This constant displays dependence on temperature [22]:(3)$$K_{H}\left(T\right)\;/\;{\mathrm{mol}\;\mathrm{kg}}^{-1}{\mathrm{bar}}^{-1}={\left(exp\left[a_{1}+a_{2}T+\frac{a_{3}}{T}+\frac{a_{4}}{T^{2}}\right]\right)}^{-1}$$

Here, a_1_ = 14.000, a_2_ = −0.013341 K^−1^, a_3_ = −558.98 K, and a_4_ = −422,577 K^2^. Thus, 1 L of a soda with P_CO_2__ = 4 bar would be expected to contain ~6 g of CO_2_ at room temperature (298.15 K, 1 kg of beverage, K_H_ = 0.0336 mol kg^−1^ bar^−1^). Upon opening such a soda, P_CO_2__ drops to the atmospheric level of CO_2_ (~0.0004 bar). Under this significantly lower pressure, essentially all the CO_2_ escapes, albeit slowly. This is fortunate for the many people that enjoy drinking carbonated beverages. Imagine if CO_2_ always escaped carbonated drinks quickly, such as what is observed when a soda bottle is opened after it has been vigorously shaken!

A number of factors contribute to the slow discharge of CO_2_ from carbonated beverages. A large activation energy barrier exists for spontaneous bubble formation, which severely limits degassing via this route [11,12]. However, CO_2_ escape can occur at the air-exposed beverage surface or at tiny, pre-existing gas cavities immersed in the bulk of the liquid [11,12,13,14,15,16,17,18,19,20]. Such gas pockets are provided by unwetted pores and pockets embedded in bits of dust, lint, and fiber that are submerged in the bulk of a beverage. These pre-existing gas bubbles serve as nucleating sites for bubble growth in what is known as Type IV bubble nucleation [11,12]. A second reason for the slow loss of CO_2_ is that the beverage/air surface does not provide a facile route for gas discharge because of its small area. Third, it is usually the case that only a few nucleation-supporting items are found in a beverage. Together, these considerations restrict the rate of CO_2_ escape from carbonated beverages. The addition of Mentos candies, however, drastically increases the rate of degassing. The surface of a Mentos candy contains enormous numbers of pits and pockets, each of which can serve as nucleation sites when submerged in a carbonated drink [3,9,10]. The maximum number of nucleation sites on a single Mentos can be grossly estimated to be between 5 and 60 million by dividing the surface area of the candy (8 cm^2^ [9]) by the circular area of the nucleation sites, which have an estimated radius of 2–7 μm [10]. Consistent with this estimate, in a previously published AFM image of the surface of a Mentos candy, about 5 pores appear in an area of 100 μm^2^ [3]. This translates to about 40 million such sites over the entire 8 cm^2^ surface of the candy. These estimates represent a maximum number of sites, given that SEM images display alternating smooth and rough regions on the Mentos surface [8]. Furthermore, the estimate assumes that all such sites are capable of actively nucleating bubbles, but the number of sites active sites likely decreases as the candy dissolves into the surrounding soda. Nevertheless, this back-of-the-envelope calculation illustrates that there are at least hundreds of thousands to millions of potential nucleation sites on a Mentos candy.

Being supersaturated in dissolved CO_2_, carbonated beverages experience a thermodynamic driving force for the escape of CO_2_ into the gas phase. However, a barrier to gas bubble formation exists due to the formation of a new gas/liquid interface. Bubbles with a radius, $r_{crit}^{*}$, that exceed a critical value can surmount this energy barrier [23,24]:(4)$$r_{crit}^{*}=\frac{-2\gamma }{∆G_{v}}$$
where *γ* is the surface tension at the gas/liquid interface and ∆*G_v_* is the free energy per molar volume associated with the process outlined in Equation (1). The radius, *r*, of a spherical bubble can also be expressed as:(5)$$r=\frac{2\gamma }{P_{in}-P_{out}}$$
where γ is the surface tension at the gas/beverage interface, $P_{in}$ is the pressure inside and $P_{out}$ is the pressure outside the bubble. From Henry’s Law $P_{in}=\frac{\left[CO_{2}\right]}{K_{H}}$, while the pressure outside the bubble is the sum of atmospheric ($P_{atm}$) and hydrostatic pressures. At the depths involved in carbonated drinks, the hydrostatic pressure is ~1% of $P_{atm}$. Thus, to a good approximation:(6)$$rr=\frac{2\gamma }{\frac{\left[CO_{2}\right]}{K_{H}}-P_{atm}}$$

In general, nucleation sites embedded within carbonated beverages are not spherical, nor are the seed bubbles that form within them. However, models of nucleation sites as cone-shaped cavities have indicated that bubbles contained therein can nucleate, grow, and detach if the radius of the cavity opening exceeds the critical radius. In such scenarios, Equation (5) is often used as a good approximation [23,25]. While the geometry of the nucleation sites on the surface of Mentos candies is not currently known, such a crevice-type model could serve as a starting point.

Using Equation (6), a critical radius of 0.5 μm can be estimated for the conditions typically found in a freshly opened carbonated beverage at room temperature ([CO_2_] = 6 g L^−1^, γ = 0.072 N m^−1^, *K_H_* = 0.0336 mol kg^−1^ bar^−1^). Cavity openings on the Mentos surface that have a radius smaller than this would not be expected to support nucleation. On the other hand, crevice openings with a radius exceeding this critical radius would support bubble nucleation, growth, and production. Therefore, Mentos candies provide a catalyst for degassing because the size of their nucleation sites (2–7 μm) exceeds the critical radius. Rearrangement of Equation (6) has provided a basis for estimating the size of the nucleation sites in Mentos [10]:(7)$$\left[CO_{2}^{*}\right]=K_{H}\left[\frac{2\gamma }{r}+P_{atm}\right]$$

In this rearranged presentation, it is assumed that the crevices on the Mentos surface have a constant pore size (r). CO_2_ concentrations that exceed the critical concentration, [CO_2_*], support bubble growth in pores of this constant size. Estimates of [CO_2_*] required to cause degassing on the surface of Mentos candies have come from observing Mentos- induced degassing rates in carbonated water at varying [CO_2_]. In these experiments, the following Equation was used to fit the kinetics observed [10]:(8)$$\frac{d}{dt}\left[CO_{2}\right]=\frac{k_{cat}K\left\{\left[CO_{2}\right]-\left[CO_{2}^{*}\right]\right\}}{1-K\left\{\left[CO_{2}\right]-\left[CO_{2}^{*}\right]\right\}}$$

In the above Equation, *k_cat_* is a kinetic constant and *K* an equilibrium constant for the diffusion of CO_2_ molecules into bubbles. In addition, in Equation (8) it is assumed that $\frac{d}{dt}\left[CO_{2}\right]=0\;$ when $\left[CO_{2}\right]<\left[CO_{2}^{*}\right]$. Equation (8) is derived from a simplified version of Brunauer–Emmett–Teller (BET) theory, an established theory that is useful for the analysis of gas adsorption in a multilayer fashion onto surfaces [26,27]. A modified version of this simplified version of BET theory is presented in the Appendix B. In BET theory, gas molecules are envisioned to bind to surface receptor sites. Additional gas molecules may adhere to gas molecules already so adsorbed onto the surface sites. Thus, BET theory requires the consideration of two distinct events as when gases interact with surfaces: gas molecules adsorbing onto bare surface sites, and gas molecules adhering to each other. The simplified version of BET theory presented here assumes that during Type IV bubble nucleation all gas molecules diffuse into bubbles. As such, only one type of event needs to be considered: individual gas molecules entering bubbles. This simpler theory provides a novel basis for evaluating the critical CO_2_ concentration that must be exceeded for a Type IV nucleation site with a radius large enough to support bubble growth, and to estimate the size of the nucleating site. This analysis has been used to ascertain that Mentos candies cause substantial degassing from water only when [CO_2_] exceeds 1.8–2.2 g L^−1^ at 25 °C [27]. When substituted into Equation (6) as [CO_2_*], these concentrations have allowed for the estimate that the pores on the Mentos candies are 2–7 μm in size [10].

How beverage additives affect various physical properties of beverages, and fountain sizes, is not entirely clear. For example, it has been argued that beverage additives which lower surface tension might enhance fountain heights by lowering the activation energy for bubble formation [3]. This suggestion contends that a lower activation energy to bubble formation might be induced by lower beverage surface tension, resulting in faster degassing kinetics and taller geysers. However, this proposal was based upon the assumption that bubbles form spontaneously within the bulk liquid of the beverage, when in fact carbonated beverages are generally agreed to degas via Type IV bubble nucleation at pre-existing cavities [11,12,13,14,15,16,17,18,19,20]. This hypothesis has further been called into question by the observation that certain beverage additives that *increase* surface tension also increase fountain heights [7]. Other physical parameters such as viscosity have been used to argue why diet sodas (lower viscosity) produce higher fountains than sugared sodas (higher viscosity) [1,5,7]. Because ethanol lowers surface tension [28] but increases viscosity [29] when added to water, examining the Mentos-induced degassing of CO_2_ from a mixture of water and ethanol could provide insight into the effects of beverage additives, beverage surface tension, and beverage viscosity on the Diet Coke and Mentos system.

To our knowledge, there exist no studies that explore the behavior of Mentos-induced degassing from alcoholic beverages. However, studies on bubble nucleation in Champagne [13,14,15,16,17,18,19,20] have shed much light on the various processes involved in the Diet Coke and Mentos system. This commonality suggests that illuminating mechanistic aspects of the Diet Coke and Mentos experiment could contribute to an understanding of various aspects of CO_2_ degassing, foaming, and bubble formation in Champagne and other carbonated beverages. These issues have provided motivation for the present studies which examine the effect of ethanol addition on Mentos-induced CO_2_ degassing from carbonated beverages and waters. The experiments presented here could also provide insights into similarities and differences in the characteristics of degassing behavior of alcoholic beverages vs. soft drinks.

## 2. Results

The Diet Coke and Mentos experiment was carried out with commercial 2 L bottles of Diet Coke or sparkling water (contains only carbonated water) prepared with increasing amounts of added ethanol by mass. In these experiments, 11 mint Mentos candies were added to each bottle, after opening, to initiate the reaction. In the samples of sparkling water to which ethanol was added, fountain heights reached as much as 6 times higher than control (no added ethanol), peaking at 1–2% ethanol, but decreasing to 3.5 times higher thereafter (Figure 1). By comparison, fountain heights in samples of ethanol-treated Diet Coke decreased slightly at low ethanol concentrations, but then increased at higher ethanol concentrations (Figure 2). Because surface tension decreases and viscosity increases with ethanol addition [28,29], these results demonstrate that neither the surface tension nor viscosity of a beverage can be linked to fountain heights in a straightforward manner. Furthermore, because alcoholic beverages generally contain ≥5% ethanol, these observations also suggest that CO_2_ escapes faster from alcoholic beverages than from soft drinks during Type IV nucleation.

To further explore the effect of ethanol on the kinetics of CO_2_ release in the Diet Coke and Mentos experiment, the kinetics of foam production was monitored at varying ethanol concentrations (Figure 3). In these experiments, 500 mL bottles of Diet Coke to which increasing amounts of ethanol were added had 1 min Mentos candy added to it, and the time-dependent foam produced was observed. Foam production was observed to occur more rapidly with increasing ethanol concentrations.

The foam volume peaked at about 5 s at 0% ethanol (Figure 3, closed circles), 4 s at 0.5% ethanol (Figure 3 open circles), and 3 s at 5% ethanol (Figure 3, closed triangles). A similar trend was observed in samples treated with 1%, 2%, and 3.5% ethanol (data in Appendix A). Furthermore, foam decay was observed to occur more rapidly at increasing ethanol concentrations. The kinetic foam data were fit to the following Equation as described previously [9]:(9)$$Y=Y_{max}\left[\frac{k_{1}k_{2}}{\left(k_{2}-k_{1}\right)\left(k_{3}-k_{1}\right)}e^{-k_{1}t}+\frac{k_{1}k_{2}}{\left(k_{1}-k_{2}\right)\left(k_{3}-k_{2}\right)}e^{-k_{2}t}+\frac{k_{1}k_{2}}{\left(k_{1}-k_{3}\right)\left(k_{2}-k_{3}\right)}e^{-k_{3}t}\right],$$

In Equation (9), *Y* is the volume of foam produced, *Y_max_* indicates the maximum amount of foam that could be produced in the absence of foam decay, *t* is time, and the *k_i_*’s are kinetic constants for the following sequential, multistep, irreversible reactions:(10)$$W\overset{k_{1}}{\to }X\overset{k_{2}}{\to }Y\overset{k_{3}}{\to }Z.$$

In all cases, R^2^ values of ≥ 0.96 were achieved (Table 1). Because the foam exhibited much turbulence over the first 1–3 s of data collection, the kinetic parameters *k*_1_ and *k*_2_ (both associated with foam production) could not be resolved. Thus, the values for these parameters consistently matched to within 0.1% of each other in each fit. Nevertheless, the numerical value of these kinetic parameters provides some quantitative insight into the rate of foam production. While not a perfect trend, higher values for *k*_1_ and *k*_2_ required for the data fits tended to increase at higher ethanol concentrations (Table 1). This observation is consistent with the faster foam production observed with increasing ethanol concentration (Figure 3). Furthermore, a solid trend of larger values for *k*_3_ (associated with foam decay) at higher ethanol concentrations was observed (Table 1), in accord with faster foam decay observed at increasing ethanol concentrations (Figure 3).

To further probe the effect of the presence of ethanol on Mentos-induced degassing from carbonated solutions, CO_2_ degassing kinetics caused by Mentos addition was monitored in solutions containing 5% ethanol in water at increasing concentrations of CO_2_. These experiments were also motivated by a previous report [10] that measured CO_2_ degassing rates from carbonated water alone. In this previously reported experiment, a plot of the rate of degassing vs. [CO_2_] was used to find [CO_2_*], which was then substituted into Equation (7) to find putative pore sizes of sites that nucleate bubbles on the Mentos candies. When Mentos were added to 5% solutions of ethanol at increasing [CO_2_], substantial degassing was only observed at concentrations higher than ~2.7 g L^−1^ (Figure 4).

The data were best fit (R^2^ = 0.99) to Equation (8) using [CO_2_*] = 2.65 ± 0.15 g L^−1^, *k_cat_* = 0.085 ± 0.009 g s^−1^, and *K* = 0.58 ± 0.01. Substitution of this critical concentration, the surface tension of 5% ethanol at 20 °C (*γ* = 0.056 N m^−1^ [28]), the recorded atmospheric pressure (P_ATM_ = 0.975 bar), and the Henry’s Law constant for CO_2_ in a 5% solution of ethanol at 20 °C ($K_{H}^{EtOH\left(aq\right)}$ = 0.0382 mol kg^−1^ bar^−1^) into Equation (8) yielded an estimated radius of 1.9 ± 0.3 μm, in agreement with previously reported estimates for the size of nucleation sites on the surface of Mentos candies [10].

Figure 4 illustrates that the Mentos-induced rate of degassing from carbonated, ethanolic solutions approaches zero as [CO_2_] approaches [CO_2_*]. Consistent with the data in Figure 4, the degassing from such solutions initially prepared at [CO_2_] >> [CO_2_*] were observed to degas rapidly at first but slow considerably over time (Figure 5). Based on this observation, it was hypothesized that [CO_2_*] could be determined by analyzing kinetic degassing data collected during a single run. This was attempted by first noting that Equation (8) can be linearized by taking its reciprocal:(11)$$\frac{1}{rate}=\frac{1}{k_{cat}K\left(\left[CO_{2}\right]-\left[CO_{2}^{*}\right]\right)}-\frac{1}{k_{cat}}$$
with $rate=\;\frac{d}{dt}\left[CO_{2}\right]$. From Equation (11) it can be seen that a plot of 1/rate vs. 1/($\left[CO_{2}\right]-\left[CO_{2}^{*}\right])$ should yield a straight line with a slope of 1/*k_cat_K* and an intercept of −1/*k_cat_*.

In this analysis, it was found that a best line of fit to the data could be achieved by varying [CO_2_*] until R^2^ was maximized. When the data displayed in Figure 5 were treated in this manner, the best linear fit (R^2^ = 0.97) was achieved when [CO_2_*] = 2.77 g L^−1^ (Figure 6). Substitution of this concentration and appropriate parameters for the conditions under which the experiment was conducted into Equation (8) resulted in a radius of 1.7 μm. The value of *k_cat_* = 0.087 g L^−1^ s^−1^ was obtained using the intercept of the linear fit and *K* = 0.32 L g^−1^ from a combination of the slope and intercept. This experiment was repeated 5 times under the same conditions of temperature (20 °C) and atmospheric pressure (0.992 bar), yielding an average critical CO_2_ concentration of 2.7 ± 0.1 g L^−1^ and an average radius of 1.9 ± 0.2 μm. The same analysis applied to degassing experiments with 5 separate runs of carbonated water (no ethanol) yielded averages of [CO_2_*] = 3.2 ± 0.4 g L^−1^ and r = 1.8 ± 0.7 μm.

## 3. Discussion

Differences in the physical properties of beverages have been offered as clues to explain why certain beverages produce higher fountains in the Diet Coke and Mentos experiment [1,3,5,7]. It has been claimed that diet sodas generate taller geysers than sugared sodas on account of the lower surface tension [3] or higher viscosity [1,5,7] of the former as compared to the latter. In the experiments presented here, it was observed that addition of ethanol to seltzer water increased fountain height (Figure 1). It was also observed that high concentrations of ethanol added to Diet Coke increased fountain heights (Figure 2). However, the viscosity of ethanol–water mixtures is higher than that of water [29]. These observations conflict with the hypothesis that beverages with lower viscosity should always outperform beverages of higher viscosity in this experiment.

It could be argued that the lower surface tension [28] imparted by ethanol addition accounts for the observed difference in heights (Figure 1 and Figure 2). However, fountain heights in seltzer water treated with 1–2% ethanol were higher than those observed at 3.5–5% ethanol (Figure 1), the latter of which have lower surface tension than the former. In addition, small amounts of ethanol (0.5–1%) added to Diet Coke slightly lowered the fountain height relative to control (Figure 2). Thus, beverages with lower surface tension do not always produce higher fountains than beverages with higher surface tension.

It is certainly the case that both surface tension and viscosity affect bubble nucleation in carbonated beverages [1,7,9,10,11,12,13,14,15,16,17,18,19,20]. It could be that the tandem of increased viscosity and decreased surface tension that accompanies ethanol addition results in competing effects with respect to fountain heights. This scenario would be consistent with fountain heights reaching a maximum at a particular ethanol concentration and decreasing thereafter, such as is observed in Figure 1. On the other hand, an opposite effect is observed in samples of Diet Coke treated with ethanol: fountain heights reach a minimum at a low ethanol concentration and increase at higher concentrations (Figure 2). The many additives in the Diet Coke likely account for this difference. Indeed, it has been noted that beverage additives impact the degree of foaming and bubble coalescence [7] in carbonated drinks, all of which could impact fountain heights. Finally, it could be that the presence of ethanol influences interactions between the beverage and the surface of the Mentos. For example, the presence of ethanol could increase the dissolution rate of some candy ingredients that might act as surfactants and enhance the kinetics of foam production. While previous studies have indicated that some ingredients in Mentos (gum Arabic and gelatin) have no effect on foaming ability [9], other ingredients in the candy could potentially play a role.

Nevertheless, the experiments presented here demonstrate that simple relationships between surface tension or viscosity and fountain heights may be difficult to establish in varied and complex beverage mixtures. Consistent with this, it has been demonstrated that beverage viscosity cannot be related to the diffusion coefficient of CO_2_ in various carbonated drinks and sodas in a straightforward manner [30]. Commercial sodas certainly contain a wide variety of solutes that could impact the degree to which foaming and fountaining occurs in a variety of ways. Thus, many aspects of the relationships between beverage viscosity, beverage surface tension, beverage additives and fountain heights in the Diet Coke and Mentos experiment still need to be explored. For now, it appears that differences in viscosity and surface tension might not sufficiently explain why diet sodas tend to form taller geysers than sugared sodas. The difference in fountain heights between diet and sugared sodas remains an open question.

Another impact of ethanol addition appears to involve faster foam generation and decay (Figure 3). Larger values of *k*_1_ and *k*_2_ were required to fit the data at higher ethanol concentration (Table 1). In addition, the kinetic constant *k*_3_, which is associated with the rate of foam decay, was noted to increase with ethanol concentration. These results once again imply that ethanol addition can impact fountain heights in competing ways. Faster foam generation is likely to be associated with rapid degassing kinetics and higher fountains. However, faster foam decay necessarily makes it more difficult to achieve large foam volumes, and may contribute to smaller fountain heights. One should be cautious when interpreting the results of these foam generation experiments (Figure 3) as being directly applicable to geyser heights observed in the original Diet Coke and Mentos experiment (Figure 1 and Figure 2). Nevertheless, these results suggest that exploring aspects of the stability of the foam generated (Figure 3) might provide insight into the heights that can be achieved in the classic experiment (Figure 1 and Figure 2).

To our knowledge, the simplified version of BET theory presented herein is a novel approach in analyzing the kinetics of Type IV bubble nucleation in carbonated beverages. In the experiments presented here, Equation (8) was used to analyze Mentos-induced degassing from carbonated aqueous solutions of 5% ethanol (Figure 4). The data collected and resulting fit allowed for an estimation of 1.9 ± 0.3 μm for the size of the nucleation sites on Mentos candies. Equation (8) was also linearized, which allowed for data from single degassing runs to be collected (Figure 5) and analyzed (Figure 6, Equation (11)). Application of this method of data collection and analysis on solutions of 5% ethanol yielded an estimation of 1.9 ± 0.2 μm for the size of nucleation sites on Mentos candies. An estimate of 1.8 ± 0.7 μm was also obtained from similar experiments conducted on solutions of carbonated water alone. These estimations for the radius of nucleating bubbles on the Mentos candies reported here were determined from experiments on liquids with quite different surface tensions and two different methods of analysis. Even so, the estimations are consistent with each other and with a previous report [10]. Interestingly, the estimates reported here are quite close to the dimensions reported for sites (0.6–2.3 μm) embedded in fibers that cause bubble nucleation in Champagne [13]. This general agreement suggests that the theory presented here could be useful in exploring Type IV degassing behavior induced by other nucleating surfaces, beverages, and solutions. These methods might also prove useful in exploring the effect that various beverage additives or mixtures of additives have on characteristics of bubble nucleation.

Results from previously reported studies conducted on bubbles in Champagne and beer [16,17,18,19] can be combined with results reported herein to make a rough prediction of the number of nucleation sites on the surface of a Mentos candy. Mentos-induced degassing of 500 mL of a Diet Coke to which ethanol was added completed in ~10 s (Figure 3). Carbonated sodas contain about ~6.0 g CO_2_ L^−1^, which is degassed by Mentos to a critical concentration of ~2.5 g L^−1^ in the presence of ethanol (Figure 3). The difference in these two concentrations is 3.5 g L^−1^, which translates to 1.75 g of CO_2_ lost from 500 mL of beverage. Using the ideal gas law, it can be shown that 1.75 g of CO_2_ corresponds to about 1 L of gaseous CO_2_ at ambient temperature and pressure. It has been noted that the flux of CO_2_ from a carbonated beverage is given by [16,17]:(12)$$\frac{dV}{dt}=Nfv$$
where *V* is the volume of CO_2_ escaping from a beverage, *N* is the number of nucleation sites inducing degassing, *f* is the frequency at which active nucleation sites produce bubbles, and *v* is the volume of an individual bubble that reaches the surface. We first take $\frac{dV}{dt}\approx \frac{∆V}{∆t}$ = 1 L per 10 s. Next, *v* in ethanol-treated Diet Coke bubbles is assumed to be the same as in Champagne (0.065 mm^3^) [18,19], and *f* for nucleation sites on Mentos to be the same as for alcoholic carbonated beverages (5–30 bubbles per second) [15,18]. Substitution of these parameters into Equation (12) results in 50,000–300,000 nucleating sites on the surface of a Mentos candy. This is substantially lower than the estimate of roughly 5–60 million pores outlined in the introduction. This discrepancy could partially be resolved by noting that the surface of the Mentos candy is not uniform, but rather contains both rough and smooth areas [8]. If the smooth areas on the surface do not contain nucleation sites, then the estimate of millions of sites could be 2–4 times too high. It could also be the case that not all pores are capable of nucleating bubbles, or that pores near one another inhibit the frequency of bubble production. Finally, there is also the issue of the candy dissolving in the beverage, which could reduce the number and sizes of sites. Regardless, further experimentation is necessary to resolve this issue.

The estimated radius of nucleating bubbles on the surface of the Mentos was not affected by the presence of ethanol. Thus, it is very likely the same types and sizes of pores on the surface of the Mentos are responsible for nucleating bubbles in both liquids. On the other hand, the critical CO_2_ concentration for degassing from water was consistently found to be ~0.5 g L^−1^ higher in water than in 5% ethanol. This observation is consistent with Equation (7), which assures that higher CO_2_ concentrations are required for degassing from liquids with higher surface tension. Further investigations could provide insight into whether it is generally true that higher critical CO_2_ concentrations correspond to liquids with higher surface tension. Because outgassing slows considerably as the CO_2_ concentration in a beverage approaches [CO_2_*] [18,19], the rate of degassing is likely to abate more quickly in beverages that display higher [CO_2_*]. Consistent with this, faster degassing kinetics (Figure 3, Table 1) and higher fountains (Figure 1 and Figure 2) are associated with increasing ethanol concentration. Further, a comparison of the critical concentrations found for Mentos-induced degassing from water (3.0 g L^−1^) and 5% ethanol (2.5 g L^−1^) indicate that an extra 0.5 g of CO_2_ are available per liter for degassing from 5% ethanol as compared to water (carbonated soft drinks generally contain ~6.0 g of CO_2_ per L). Beverages with lower [CO_2_*] have more available CO_2_ to degas, promoting faster kinetics and higher fountains. These factors could contribute to an explanation of the higher fountains observed in ethanol-containing samples as compared to their controls.

## 4. Materials and Methods

### 4.1. Measurement of Fountain Heights

Two-liter bottles of either Diet Coke or commercial sparkling water (contains only carbonated water) with the same expiration date were purchased from the same store. The bottles were opened, and 100 g of the beverage was removed. Immediately after, additions of 200 proof ethanol and water were made such that the same mass of fluid (to within 1 g) was added to all samples. After resealing, each bottle was well mixed and allowed to rest undisturbed at room temperature for several hours. After this, the bottles were placed on a flat surface and 11 Mint Mentos candies were added into a bottle. At least three trials were run for samples at each ethanol concentration. A video of the fountain produced was recorded and analyzed for height as previously described [7,9].

### 4.2. Foam Volume Kinetics

Bottles of Diet Coke (500 mL) from the same six-pack and purchased from the same store were opened and additions of 200 proof ethanol and water were made. The same mass of fluid (to within 1 g) was added to all samples. Bottles were resealed, mixed well, and allowed to rest undisturbed until they reached room temperature. To carry out the tests, a bottle was opened and a home-built foam collector [9,10] was securely fastened to the top of the bottle (Figure 7). A running stopwatch was placed next to the assembly, a Mentos candy was added into the foam collector, and a video of the foam release was recorded and later analyzed to collect foam volume vs. time data as previously described [9,10]. At least three trials were run for samples at each ethanol concentration. The resulting data were fit to Equation (9) using a least squares analysis via the Solver function in Excel. Attempts were made to fit the data using the equation for a two-step, irreversible mechanism ($X\overset{k_{1}}{\to }Y\overset{k_{2}}{\to }Z$). However, such fits were not used because they routinely yielded R^2^ values lower than 0.9.

### 4.3. Kinetics of CO_2_ Release from Carbonated Water and Aqueous Ethanol Solutions

The atmospheric pressure in the lab was recorded. Empty 355 mL soda bottles were rinsed three times with deionized water, filled with 310 g of a solution of 5% ethanol by mass in water at 20 °C, and placed on an Ohaus Scout Pro balance (600 g capacity and 0.01 g precision) interfaced with a computer running LoggerPro data acquisition software. After the balance was tared, the bottle was removed and sealed, and then CO_2_ was added using a home carbonation system (Fizz Giz, High Point, NC, USA). After the bottle rested undisturbed for two minutes, it was opened and placed on the balance to find the mass of CO_2_ added. Next, one mint Mentos candy was added to initiate the degassing reaction. Initial rates of degassing were determined by finding the slope of the resulting mass vs. time data recorded in LoggerPro. The data set collected was fit to Equation (8) (assuming $\frac{d}{dt}\left[CO_{2}\right]=0$ for $\left[CO_{2}\right]<\left[CO_{2}^{*}\right]$) using a least squares analysis and [CO_2_*], *k_cat_*, and *K* as variable parameters. The Solver function and Solver Statistics add-in [31] were used in Excel to determine the values and errors of these variables. The critical CO_2_ concentrations determined by this method were substituted into Equation (6) to find r, assumed to be the radius of pore openings on the Mentos candies responsible for the initial rate of degassing. The Henry’s Law constant for CO_2_ in water, $K_{H}^{H_{2}O}$, used in Equation (6) was found using Equation (3). The Henry’s Law constant for CO_2_ in a solution of 5% by mass ethanol,$\;K_{H}^{EtOH\left(aq\right)}$, was estimated to be the same as $K_{H}^{H_{2}O}$ by noting that the Henry’s Law constants for water and dilute solutions of aqueous ethanol are good approximations of one another [32,33]. For single runs analyzed using the reciprocal plot (Equation (11)), samples were treated in the same way with the following exceptions. Empty 591 mL soda bottles were rinsed three times with deionized water and filled with 500 g of either water or 5% by mass ethanol at 20 °C. The mass of the Mentos candy was recorded prior to adding it to initiate the reaction, and the mass of CO_2_ was taken by difference. After opening a bottle, a candy was added, and the time-dependent mass of CO_2_ was recorded. The rate of degassing at time *t* was determined by taking the slope of the five data points from time *t* to time *t* + 4. Data were plotted as 1/rate vs. 1/($\left[CO_{2}\right]-\left[CO_{2}^{*}\right])$ and the R^2^ value for the resulting plot was calculated in Excel. The data plotted as such were best fit to a line using the Solver function in Excel, varying [CO_2_*] to make R^2^ as close to 1 as possible. Data from the first 2–3 s after adding the candy were often excluded from the inverse linear plots due to instability in mass measurements immediately after adding the candy. Because *k_cat_* should be a positive parameter, fits that (on occasion) resulted in a positive y-intercept (which is equal to −1/*k_cat_*) were discarded. Because the sensitivity of the balance used was 0.01 g, only rates that exceeded 0.01 g s^−1^ were included for analysis in the reciprocal plot.

## 5. Conclusions

A variety of methods were used to explore the effect of ethanol on the kinetics of degassing and bubble nucleation in the Diet Coke and Mentos experiment. The presence of ethanol was found to increase fountain heights (Figure 1 and Figure 2), CO_2_ degassing kinetics (Figure 3), and kinetics of foam decay (Figure 3). It was proposed that ethanol addition causes a lower critical CO_2_ concentration in carbonated waters, which promotes a faster and greater quantity of CO_2_ release. It is possible that ethanol addition supports higher fountain heights by decreasing beverage surface tension, but also impedes fountain heights by increasing viscosity. Contrasting effects of ethanol addition were also evident in the kinetics of foam production and decay. Foam production and decay were both observed to accelerate with increasing ethanol concentration (Figure 3, Table 1). The former effect likely contributes to enhanced fountain heights, while the latter effect could contribute to smaller geysers. On the basis of the disparate effects of ethanol addition on the experiments observed herein, it is argued that care should be taken when ascertaining the effects of surface tension, viscosity, and beverage additives in the Coke and Mentos experiment.

A novel, simplified version of BET theory provides two equations that were useful in analyzing varied experiments involving the Mentos-induced degassing kinetics from both carbonated water and carbonated solutions of ethanol. The resulting methods of data collection and analysis employed were useful in predicting the radius of pore openings acting as nucleating sites on the surface of the Mentos candies. The presence of ethanol was not found to impact these estimated sizes, suggesting that the method of analysis may be a useful probe of the effect of other nucleating agents in the Diet Coke and Mentos experiment and bubble nucleation in general. With the use of more sophisticated equipment, higher quality kinetic data could be collected than is reported here. Analysis of such data might have potential to probe the effect of various beverage additives on bubble nucleation through comparison of the fitting parameters *k_cat_* and *K* in Equations (8) and (11).

## Figures and Tables

**Figure 1 molecules-26-01691-f001:**
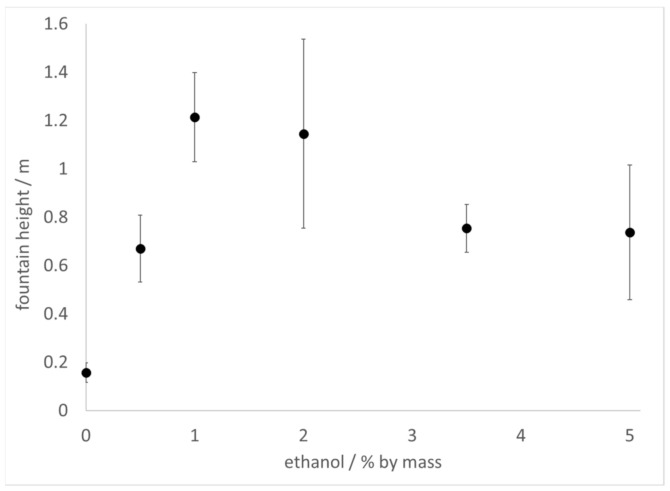
Fountain heights achieved when 2 L of sparkling water is treated with increasing amounts of ethanol as indicated in the text and then 11 mint Mentos are added at 20–21 °C. Error bars represent one standard deviation.

**Figure 2 molecules-26-01691-f002:**
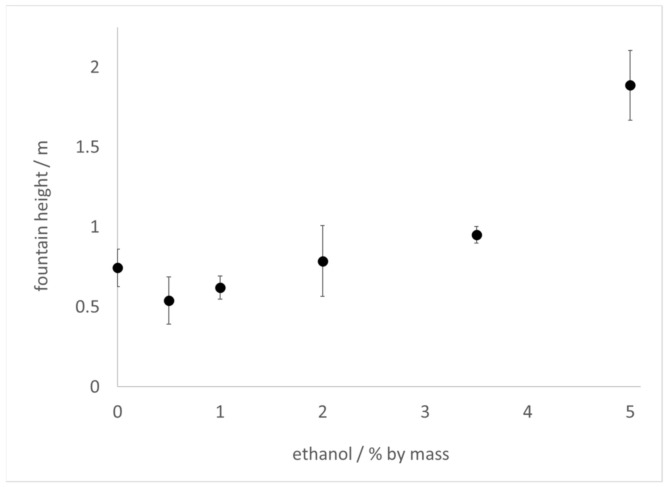
Fountain heights achieved when 2 L of Diet Coke is treated with increasing amounts of ethanol as indicated in the text and then 11 mint Mentos are added at 20–21 °C. Error bars represent one standard deviation.

**Figure 3 molecules-26-01691-f003:**
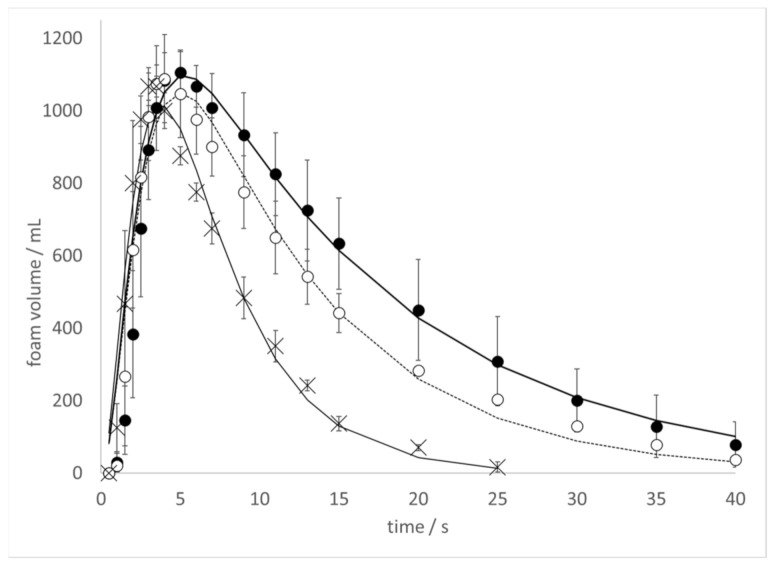
Kinetics of foam production when 1 min Mentos candy is added to 500 mL of Diet Coke at 20–21 °C as described in the text. (**Closed circles**) Control samples to which water was added; (**open circles**) samples to which 0.5 %; and (**X’s**) 5% ethanol by mass was added. Lines represent fits to the data using parameters listed in Table 1. Error bars represent one standard deviation.

**Figure 4 molecules-26-01691-f004:**
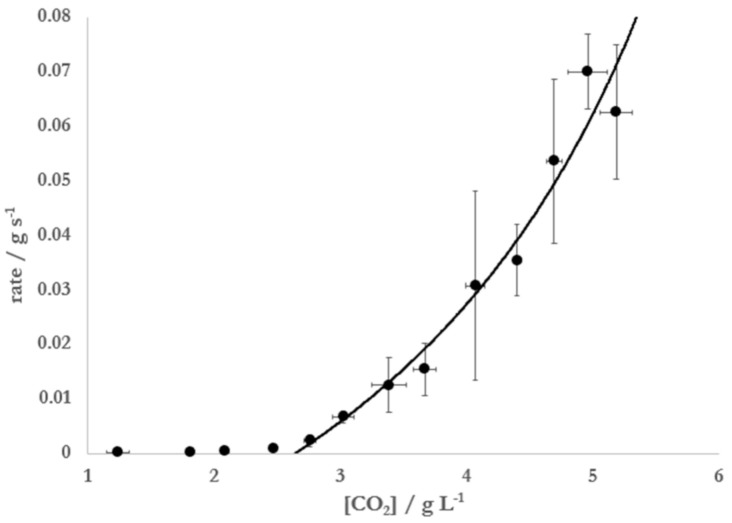
Rates of degassing from 5% by mass solutions of ethanol in water at 20 °C and P_atm_ = 0.975 bar. Degassing was initiated by addition of a single Mentos candy. Error bars represent 95% confidence interval.

**Figure 5 molecules-26-01691-f005:**
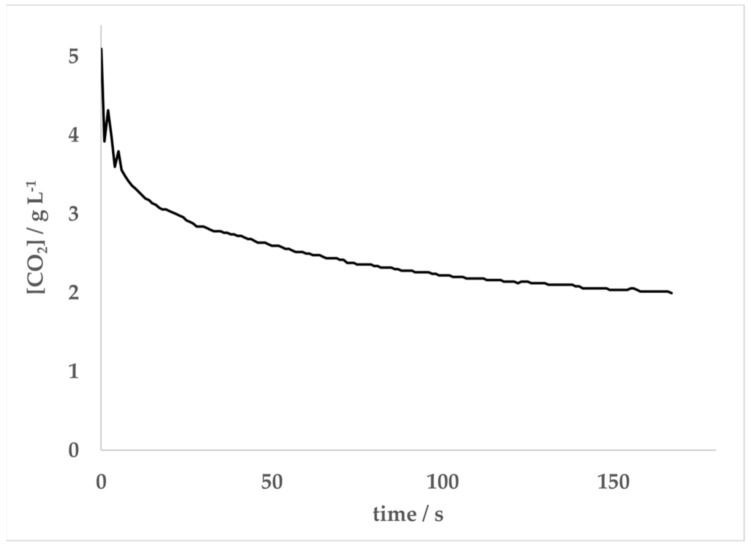
Time-dependent kinetics of CO_2_ escape from 500 g of a solution containing 4.84 g L^−1^ of CO_2_ and 5% by mass ethanol at 20 °C and P_atm_ = 0.992 bar. The reaction was initiated by the addition of a single Mentos candy at *t* = 0 s.

**Figure 6 molecules-26-01691-f006:**
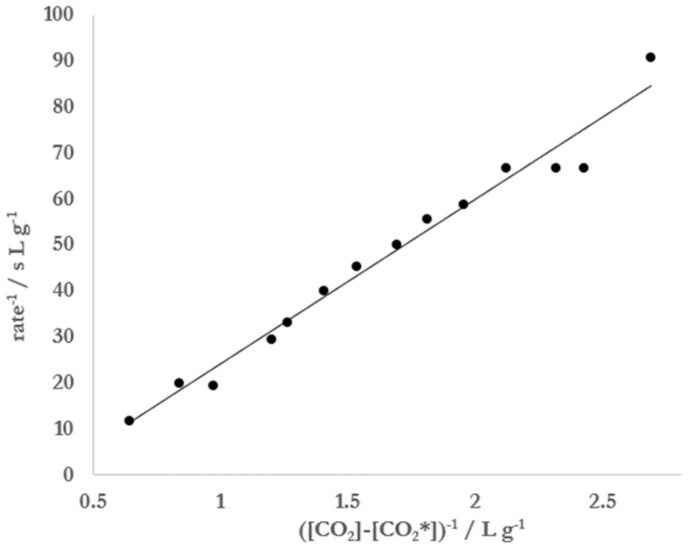
Inverse of the rate of degassing vs. the inverse of ([CO_2_] − [CO_2_*]) for a 5% by mass solution of ethanol in water. The data presented here represent data from Figure 4 from t = 3 to 17 s. The best linear fit to these data was achieved (R^2^ = 0.97) when [CO_2_*] = 2.77 g L^−1^.

**Figure 7 molecules-26-01691-f007:**
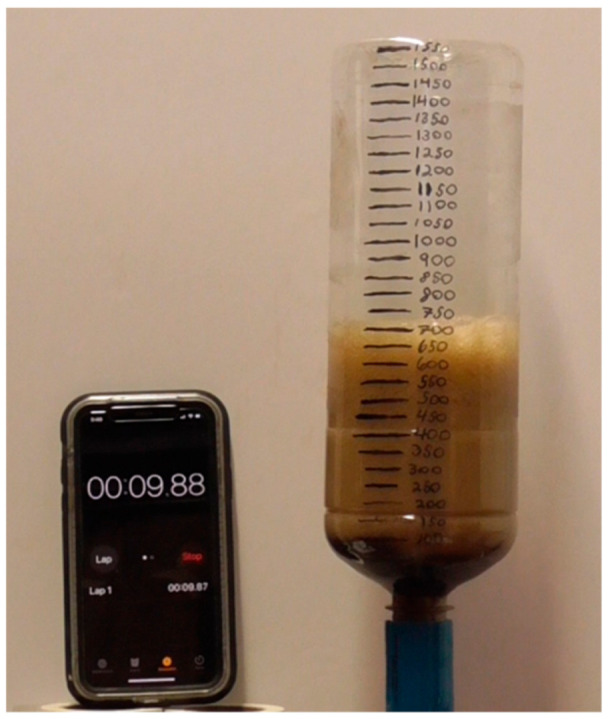
Image of home-built foam collector pictured 4.88 s after dropping a single Mentos candy into a sample prepared from 500 mL of Diet Coke at 20 °C.

**Table 1 molecules-26-01691-t001:** Parameters used to fit kinetic foam data to Equation (8).

% Ethanol by Mass	*Y_max_*	*k* _1_	*k* _2_	*k* _3_	R^2^
0	1630 mL	0.60 s^−1^	0.60 s^−1^	0.083 s^−1^	0.97
0.5%	1600 mL	0.72 s^−1^	0.72 s^−1^	0.107 s^−1^	0.96
1%	1670 mL	0.81 s^−1^	0.81 s^−1^	0.110 s^−1^	0.97
2%	1480 mL	0.80 s^−1^	0.80 s^−1^	0.145 s^−1^	0.97
3.5%	1430 mL	0.93 s^−1^	0.93 s^−1^	0.170 s^−1^	0.97
5%	1930 mL	0.91 s^−1^	0.91 s^−1^	0.175 s^−1^	0.96

## Data Availability

Data is contained within the article or Appendix A.

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
