# Peer review of "Ethanol as a Probe for the Mechanism of Bubble Nucleation in the Diet Coke and Mentos Experiment"

_molecules, 2021, doi:10.3390/molecules26061691_

Round 1

Reviewer 1 Report

Overall, the discussion seemed to ignore some. I think that the surface of Mentos must include a surfactant, and the intrinsic problem derived from the effect by surfactant had been claimed in the previous study (e.g., ref. 3). The addition of ethanol might just facilitate the dissolution rate of surfactant or wettability, thus enhancing the fountain. But, the author just discussed the surface tension at the liquid-air interface and no interfacial science between the Mentos and solution has been discussed. Thus, I cannot readily receive that the “current descriptions of the effects of surface tension are overstated” as written in the abstract.

In addition, the fact that only 5 wt% of ethanol condition was used has lowered the generality of the product obtained in this article. The author has mentioned the effect of surface tension that “higher CO2 concentrations are required for degassing from liquids with higher surface tension”, but I wonder how generality is included for the assessment? Though the authors have reported on the sugar alcohol [ref. 7], does it generalized considering those previous data and other alcohols? I think the document is interesting as reading material, but I don’t think that the article includes sufficient “Strategy for the Chemical Analysis” as the original article in the present form. I would the authors to empathize where the scientific novelty is. If the use of 5 wt% ethanol was a novel issue, getting novel scientific insight must be difficult. More variables must be needed.

Author Response

Dear Reviewer,

Thank you very much for taking the time to go over our work. Your help is very much appreciated. We welcome any further suggestions you might have for improvements to the manuscript.

Overall, the discussion seemed to ignore some. I think that the surface of Mentos must include a surfactant, and the intrinsic problem derived from the effect by surfactant had been claimed in the previous study (e.g., ref. 3). The addition of ethanol might just facilitate the dissolution rate of surfactant or wettability, thus enhancing the fountain. But, the author just discussed the surface tension at the liquid-air interface and no interfacial science between the Mentos and solution has been discussed. Thus, I cannot readily receive that the “current descriptions of the effects of surface tension are overstated” as written in the abstract.

Thank you for pointing out that the phrase “current descriptions of the effects of surface tension are overstated” is an overreach on our part. We have therefore removed this phrase from the abstract.

Our experiments were intended to show that comparisons of beverage surface tension (or beverage viscosity) alone cannot explain differences in fountain height. This is indeed illustrated in the results in Figure 1.

We thank you for pointing out that the ethanol might facilitate dissolution rate of surfactants on the surface of the Mentos candies. Towards this end, we have added the following discussion to lines 263-274 of an updated version of the manuscript:

It could be that the lower surface tension [28] imparted by ethanol addition accounts for the observed difference in heights (Figure 1). On the other hand, the height of the fountains observed in control samples of Diet Coke (g ~ 0.072 N m-1) were almost twice as high (Figure 1) as those in the ethanol-water mixture (g ~ 0.056 N m-1). Thus, it is not always the case that beverages with lower surface tension produce higher fountains than beverages with higher surface tension. It could be that the presence of ethanol impacts interactions between the beverage and the surface of the Mentos. For example, the presence of ethanol could increase the dissolution rate of certain candy ingredients that might act as surfactants and enhance the kinetics of foam production. While previous studies have indicated that some ingredients in Mentos (gum Arabic and gelatin) appear to have no effect on foaming ability [9], other ingredients in the candy could potentially play a role.

In addition, the fact that only 5 wt% of ethanol condition was used has lowered the generality of the product obtained in this article. The author has mentioned the effect of surface tension that “higher CO2 concentrations are required for degassing from liquids with higher surface tension”, but I wonder how generality is included for the assessment?

Thank you for alerting us to this. The full context of our use of the phrase “higher CO2 concentrations are required for degassing from liquids with higher surface tension” is intended to show consistency with Equation 7, but not a statement of generality (lines 371-373 of an updated manuscript):

This observation is consistent with Equation 7, which assures that higher CO2 concentrations are required for degassing from liquids with higher surface tension.

We have added the following sentence to line to make this point a bit more clear (lines 373-375 of updated manuscript):

Further investigations could provide insight into whether it is generally true that higher critical CO2 concentrations correspond to liquids with higher surface tension.

Though the authors have reported on the sugar alcohol [ref. 7], does it generalized considering those previous data and other alcohols?

Please note that we have previously reported on similar experiments using various alcohols and alcohol concentrations (see the supporting information associated with reference 7). We have therefore referenced these experiments as indicated below (lines 260-268 of updated manuscript):

Consistent with previously reported experiments with alcohols [30], ethanol addition enhanced degassing kinetics and fountain heights in the Coke and Mentos experiment (Figures 1-2). It could be that the lower surface tension [28] imparted by ethanol addition accounts for the observed difference in heights (Figure 1). On the other hand, the height of the fountains observed in control samples of Diet Coke (g ~ 0.072 N m-1) were almost twice as high (Figure 1) as those in the ethanol-water mixture (g ~ 0.056 N m-1). Thus, it is not always the case that beverages with lower surface tension produce higher fountains than beverages with higher surface tension.

I think the document is interesting as reading material, but I don’t think that the article includes sufficient “Strategy for the Chemical Analysis” as the original article in the present form. I would the authors to empathize where the scientific novelty is. If the use of 5 wt% ethanol was a novel issue, getting novel scientific insight must be difficult. More variables must be needed.

To our knowledge, application of BET theory to bubble nucleation is a novel approach. We have tried to emphasize this approach at several places in the manuscript (lines 316-327, line 458 in the conclusion, and the appendix of the updated manuscript).

Thank you once again for your time and expertise,

Tom Kuntzleman

Reviewer 2 Report

The authors mention that beverage additives have been previously shown have synergistic effects on foam (J. Agric. Food Chem. 2011, 59, 3168-3179). A brief discussion on the types of beverage additives have previously been observed to produce the synergistic effects in foam formation would aid the reader in understanding how these synergistic effects might impact foam formation in the Diet Coke and Mentos system.

A previous study on the Diet Coke and Mentos experiment by Coffey (Am. J. Phys. 2008, 76, 551-557) used atomic force microscopy and scanning electron microscopy to characterize the surface of the Mentos candy. Are the estimates for the number of nucleation sites that are calculated by the authors consistent with the results of previous observations of the candy surface?

The authors observe that foam decays faster in samples of Diet Coke-ethanol than in the control experiments (larger values of k3), and they note that the faster decay makes it more difficult to achieve large foam volumes. The authors also show in Figure 2 that both the ethanol treated Diet Coke and the controls generate similar volumes of foam. Could the similarities in foam volume be the result of a lower value for [CO2*] for ethanol treated Diet Coke that results in more foam being generated to replace the more quickly decaying foam in the Diet Coke-ethanol samples? In other words, it is possible that the ethanol treated Diet Coke really generates more foam, but because of the faster foam decay it appears to generate an equal volume of foam compared to the control?
In heading of section 4.3 (line 378) there is a typo that has a lowercase O in the formula for CO2

Author Response

Dear Reviewer,

Thank you very much for taking the time to go over our work. Your help is very much appreciated. We covet any further suggestions you might have for improvements to the manuscript.

The authors mention that beverage additives have been previously shown have synergistic effects on foam (J. Agric. Food Chem. 2011, 59, 3168-3179). A brief discussion on the types of beverage additives have previously been observed to produce the synergistic effects in foam formation would aid the reader in understanding how these synergistic effects might impact foam formation in the Diet Coke and Mentos system.

Thank you for asking us to check on this. The reference we supplied indicates increased foaming ability via a synergy between high molecular weight (> 12 kDa) compounds and hydrophobic, low molecular weight (> 1 kDa) compounds. Because Diet Coke contains no high molecular weight compounds, such a synergy is unlikely to provide insight into the foaming that occurs in the Coke and Mentos system. Because of this, we have removed this reference and the mention in the manuscript that synergy between compounds might affect foaming in the Coke and Mentos system. 

A previous study on the Diet Coke and Mentos experiment by Coffey (Am. J. Phys. 2008, 76, 551-557) used atomic force microscopy and scanning electron microscopy to characterize the surface of the Mentos candy. Are the estimates for the number of nucleation sites that are calculated by the authors consistent with the results of previous observations of the candy surface?

Thank you for this suggestion. It appears there is a discrepancy between the estimates of the maximum number of nucleating sites on the Mentos candies and the number of sites that appear to be generating bubbles. We outline these estimates and discrepancies in lines 67-78 of the Introduction and lines 357-366 of the Discussion section of an updated version of the manuscript. This text is indicated below:

The maximum number of nucleation sites on a single Mentos can be grossly estimated to be between 5 and 60 million by dividing the surface area of the candy (8 cm2 [9]) by the circular area of the nucleation sites, which have an estimated radius of 2–7 mm [10]. Consistent with this estimate, in a previously published AFM image of the surface of a Mentos candy, about 5 pores appear in an area of 100 mm2 [3]. This translates to about 40 million such sites over the entire 8cm2 surface of the candy. These estimates represent a maximum number of sites, given that SEM images display alternating smooth and rough regions on the Mentos surface [8]. Furthermore, the estimate assumes that all such sites are capable of actively nucleating bubbles, but the number of sites active sites likely decreases as the candy dissolves into the surrounding soda. Nevertheless, this back-of-the-envelope calculation illustrates that there are at least hundreds of thousands to millions of potential nucleation sites on a Mentos candy.

Substitution of these parameters into Equation 12 results in 50,000–300,000 nucleating sites on the surface of a Mentos candy. This is substantially lower than the estimate of roughly 5-60 million pores outlined in the introduction. This discrepancy could partially be resolved by noting that the surface of the Mentos candy is not uniform, but rather contains both rough and smooth areas [8]. If the smooth areas on the surface do not contain nucleation sites, then the estimate of millions of sites could be 2-4 times too high. It could also be the case that not all pores are capable of nucleating bubbles, or that pores near one another inhibit bubble production and frequency. Regardless, further experimentation is necessary to resolve this issue. 

The authors observe that foam decays faster in samples of Diet Coke-ethanol than in the control experiments (larger values of k3), and they note that the faster decay makes it more difficult to achieve large foam volumes. The authors also show in Figure 2 that both the ethanol treated Diet Coke and the controls generate similar volumes of foam. Could the similarities in foam volume be the result of a lower value for [CO2*] for ethanol treated Diet Coke that results in more foam being generated to replace the more quickly decaying foam in the Diet Coke-ethanol samples? In other words, it is possible that the ethanol treated Diet Coke really generates more foam, but because of the faster foam decay it appears to generate an equal volume of foam compared to the control?

Thank you for these comments. These observations can be made quantitative by comparing the values of Ymax (interpreted as the maximum foam volume in the absence of foam decay) required to fit the data in Figure 2: Ymax for ethanol treated samples is a little over 10% higher than Ymax for controls. We note this in lines 189-191 of an updated version of the manuscript:

A higher value of Ymax was also required to fit the data from the ethanol treated samples as compared to controls (Table 1).

As noted, these issues are also fully discussed in what is now lines 296-345 of an updated version of the manuscript.

In heading of section 4.3 (line 378) there is a typo that has a lowercase O in the formula for CO2

Thank you for catching this error. We have corrected this (line 417 in the updated manuscript).

Thank you again for your time and expertise.

With appreciation,

Tom Kuntzleman

Reviewer 3 Report

While I do not dispute the major finding of the paper, there are a few points that the authors should elaborate on.

Some of the foundational introductory material has to be addressed. Mentos and Coke is a popular demonstration and it does illustrate some interesting physical concepts in a dramatic, compelling way. That said, we’ve moved beyond the “gee whiz” stage now.

Equation 4 in the paper is the starting point. That equation is a re-statement of Laplace’s Equation, which is a condition for mechanical stability. See pgs. 3-6 in Dufour and Defay, for example. The radius in that equation is not the same radius as the critical radius needed for nucleation of a new phase from a supersaturated state of an old one. See Chapter 11 in Dufour and Defay for a discussion.

In essence, a statement of the Laplace pressure is not sufficient for a discussion of nucleation in this system.

Its the imbalance in the chemical potentials of CO2 in the gas phase in the bubble and the CO2 in the aqueous phase that drives the diffusion of CO2 into the bubble, which results in bubble growth and eventual detachment of the bubble such that it makes its way to the surface. Equality of those chemical potentials is where the critical radius of the bubble from the perspective of nucleation theory arises. That must be smaller than the sites on the Mentos, which is why you have the bubble production in the system. Note that the surface tension also enters the expression for the critical radius derived from classical nucleation theory.

The process in champagne seems to have been well studied, as the authors of this paper note. From Liger-Belair (2005), “In the case of liquids with low supersaturating ratios, such as sparkling wines and carbonated beverages in general, bubbles need pre-existing gas cavities larger than a critical size to overcome the energy barrier and grow freely (11, 12).” [11 and 12 are references in that paper.] The Mentos must be providing those cavities. But the Mentos must be dissolving, meaning that the cavities are continuously changing size.

One final note... the change in surface tension and viscosity upon adding ethanol isn’t the only possible change in the system. It could change the chemical potential of the CO2 in the aqueous phase, which would change the rate at which CO2 is diffusing. That might be measured simply by measuring the mass as CO2 diffuses out when opened. (Not asking that this be done for this paper, just mentioning it as an avenue for exploration.)

Dufour, L. and Defay, R., 1963. Thermodynamics of Clouds, International Geophysics Series. Academic Press, New York and London, 1965.

Liger-Belair, G., 2005. The physics and chemistry behind the bubbling properties of champagne and sparkling wines: a state-of-the-art review. Journal of Agricultural and Food Chemistry, 53(8), pp.2788-2802.

Author Response

Dear Reviewer,

Do not hesitate to contact us with questions or concerns.

Thank you for going over our work.

Tom Kuntzleman

Round 2

Reviewer 1 Report

The author used ethanol to mention the less importance of the surface tension (or viscosity) of the solution in the experiment. But, in other discussions, no quantitative scientific argument in terms of the effect of ethanol was discussed. I think that the authors should gather the data at the different EtOH concentrations (i.e. 10 %) to support the author’s proposal and not to use 5% ethanol condition only to overturn the surface-tension-based concept. In addition, for my idea, the description of the aim of this article is blurry or abstract. I think that the author should more specifically describe what the author has newly done, including an introduction of BET theory.

Author Response

Reviewer #1:

Dear Reviewer,

Thank you for taking the time once again to go over our work. Your time and suggestions are greatly appreciated.

The author used ethanol to mention the less importance of the surface tension (or viscosity) of the solution in the experiment. But, in other discussions, no quantitative scientific argument in terms of the effect of ethanol was discussed. I think that the authors should gather the data at the different EtOH concentrations (i.e. 10 %) to support the author’s proposal and not to use 5% ethanol condition only to overturn the surface-tension-based concept.

Thank you for suggesting these further experiments. We have gone back and repeated experiments at a variety of ethanol concentrations that range from 0.5% to 5% ethanol by mass. The results of these experiments are presented in Figures 1-3 of the updated version of the manuscript. These experiments have been very useful in allowing us to more fully describe the effects of ethanol addition, beverage surface tension, and beverage viscosity on the Coke and Mentos fountain heights. We invite you to read lines 170-231 of the results section, lines 290-352 in the discussion section, and lines 475-490 in the conclusion section of the updated manuscript, which cover these new experiments in more detail.  

In addition, for my idea, the description of the aim of this article is blurry or abstract. I think that the author should more specifically describe what the author has newly done, including an introduction of BET theory.

We have made several changes to the document (such as indicated above – lines 170-231, lines 290-352, and lines 475-490) to describe more fully the aims of the article, what is novel about the work, and to introduce BET theory. Please also consider the following excerpt that has been added to the Introduction section:

 Equation 8 is derived from a simplified version of Brunauer-Emmett-Teller (BET) theory, an established theory that is useful for the analysis of gas adsorption in a multi-layer fashion onto surfaces [26, 27]. A modified version of this simplified version of BET theory is presented in the appendix. In BET theory, gas molecules are envisioned to bind to surface receptor sites. Additional gas molecules may adhere to gas molecules already so adsorbed onto the surface sites. Thus, BET theory requires the consideration of two distinct events as when gases interact with surfaces: gas molecules adsorbing onto bare surface sites, and gas molecules adhering to each other. The simplified version of BET theory presented here assumes that during Type IV bubble nucleation all gas molecules diffuse into bubbles. As such, only one type of event needs to be considered: individual gas molecules entering bubbles. This simpler theory provides a novel basis for evaluating the critical CO2 concentration that must be exceeded for a Type IV nucleation sites with a radius large enough to support bubble growth, and to estimate the size of the nucleating site.

Thank you again for your time,

Tom Kuntzleman

Reviewer 3 Report

The authors have addressed my comments in a satisfactory way.

The introductory material that they have added will, I think, be enough to alert the reader that what’s being considered here is from the perspective of mechanical stability, but that bubble nucleation is, in general, driven by difference in chemical potential.

I have a few additional points for consideration. A couple of these I missed in my first review for which I apologize. There’s nothing in these comments that I feel the authors must address for publication.

Discussion around equations 4 and 5.

Equation 4 is the equation governing nucleation. That is the critical radius for a bubble to be in equilibrium with the CO2 dissolved in the liquid.

Equation 5 is the equation showing the critical radius for mechanical stability. Those two radii don’t have to be the same.

This could easily be denoted by adding a subscript to r in both equations.

Lines 149-158

The sentences beginning “This commonality suggests that…” and the one beginning “It is hoped that such experiments…” are redundant. One could be deleted.

Figure 1

This information would be better displayed as a table. Histograms are appropriate when the width of the bin matters. Here the width of the bar is irrelevant. And the order that they are displayed is also arbitrary.
